# Association of the Mannose-Binding Lectin 2 BB Genotype with COVID-19-Related Mortality

**DOI:** 10.3390/life13020382

**Published:** 2023-01-30

**Authors:** Yasuyo Kashiwagi, Shinji Suzuki, Ryo Takahashi, Gaku Yamanaka, Yuji Hirai, Hisashi Kawashima

**Affiliations:** 1Department of Pediatrics and Adolescent Medicine, Tokyo Medical University, Tokyo 160-0023, Japan; 2Department of Infectious Diseases, Tokyo Medical University Hachioji Medical Center, Tokyo 160-0023, Japan; 3Kohsei Chuo General Hospital, Tokyo 160-0023, Japan

**Keywords:** BB genotype, coronavirus disease 2019 (COVID-19), mannose-binding lectin (MBL)

## Abstract

**Simple Summary:**

In Japan, there have been few reports to date regarding the association between mannose-binding lectin, as an important part of innate immune defenses, and COVID-19. It has been demonstrated that the *mannose-binding lectin 2* gene B variant at codon 54 (*rs1800450*) is associated with variabilities in the clinical course of COVID-19. We aimed to investigate how the level of serum mannose-binding lectin and the codon 54 variant of mannose-binding lectin (*rs1800450*) affect the disease severity of COVID-19. Binary logistic regression analysis to identify predisposing factors for severe COVID-19 symptoms demonstrated that patients with the BB genotype have a higher risk of death from COVID-19 than those with the AA and AB genotype. Our results quantitatively demonstrated that the BB genotype might be a factor associated with death from COVID-19. Mannose-binding lectin might be a key factor in establishing new treatment strategies and developing novel therapeutic drugs.

**Abstract:**

Mannose-binding lectin (MBL) is crucial in first-line immune defenses. There are still many unknown factors regarding the mechanisms causing variability in the clinical course of coronavirus disease 2019 (COVID-19). In Japan, there have been few reports to date regarding the association between MBL and COVID-19. It has been demonstrated that the *MBL2* gene B variant at codon 54 (*rs1800450*) is associated with variabilities in the clinical course of COVID-19. We aimed to investigate how the level of serum MBL and the codon 54 variant of MBL (*rs1800450*) affect the disease severity of COVID-19. A total of 59 patients from the fourth wave and 49 patients from the fifth wave in Japan were analyzed based on serum MBL levels using ELISA and the genotype of *MBL2* codon 54 using PCR reaction. There was no significant association between serum MBL levels and age. *MBL2* genotype was independent of age, there was no significant difference in different COVID-19 severities, MBL genotypes, and serum MBL levels. Binary logistic regression analysis to identify predisposing factors for severe COVID-19 symptoms demonstrated that patients with the BB genotype had a higher risk of death from COVID-19. Our results quantitatively demonstrated that the BB genotype might be a factor associated with death from COVID-19.

## 1. Introduction

Mannose-binding lectin (MBL) is crucial to the first-line immune defenses against several important pathogens, including severe acute respiratory syndrome coronavirus 2 (SARS-CoV-2). MBL binds oligosaccharides on the surface of microorganisms, and it activates mannose-associated serine proteases (MASPs).

MBL and MASPs together mediate coagulation factor-like activities, including thrombin-like activity. They have direct opsonization and virucidal activity and activate the lectin pathway of the complement system. The involvement of MBL in immune protection and disease complications has been demonstrated. MBL deficiency leads to disseminated intravascular coagulation and organ failure and increases susceptibility to infection, and the mechanisms involved include reduced opsonophagocytic killing and reduced activation of the lectin complement pathway [1].

MBL is encoded by the *MBL2* gene, localized on the long arm of chromosome 10 and comprising four exons. MBL insufficiency or dysfunction is caused by three structural variants in exon 1. One hypothesis states that these *MBL2* gene variants result in susceptibility to various infectious and autoimmune diseases, such as rheumatoid arthritis [2], Kawasaki disease [3,4], recurrent vulvovaginal candidiasis [5], respiratory distress syndrome in preterm infants [6], hepatitis B infection [7], severe acute respiratory syndrome coronavirus infection [8], etc. However, whether these *MBL2* gene variants are really associated with susceptibility to various diseases remains unknown and controversial.

Three variant alleles of the *MBL2* gene (B, C, and D) in exon 1 result from codons 54, 57, and 52, respectively. The wild type is referred to as allele A. In the B allele, codon 54 is changed from GGC to GAC, causing an amino acid replacement from glycine to aspartic acid (p. Gly54Asp) [9]. Medetalibeyoglu et al. [10] investigated whether the *MBL2* gene B variant at codon 54 (*rs1800450*) is associated with variabilities in the clinical course of coronavirus disease 2019 (COVID-19), and they found that the B variant is associated with lower MBL levels and a higher risk of a more severe clinical course of COVID-19.

COVID-19 is an infectious disease caused by SARS-CoV-2. The disease was announced by the World Health Organization on March 11, 2020, as a global public health emergency and pandemic. The COVID-19 pandemic has spread rapidly and widely around the world, and it has resulted in high morbidity and mortality.

There are still many unknown factors regarding the mechanisms that cause variability in the clinical course of COVID-19. It is important to clarify the mechanisms that underly variabilities in the clinical course of COVID-19 in order to establish new treatment strategies and develop novel therapeutic drugs. For diseases other than COVID-19, novel therapeutic approaches, such as complement-targeting therapies, are currently being studied [11,12]. We aimed to investigate how the level of serum MBL and the codon 54 variant of the *MBL2* gene (*rs1800450*) affect the disease severity of COVID-19.

## 2. Materials and Methods

### 2.1. Patients and Study Design

This study was conducted in accordance with the Declaration of Helsinki and performed in accordance with the ethics standards of the Ethics Committee of Tokyo Medical University (study approval no.: T2020-0052). The study included patients who were diagnosed as having COVID-19 through polymerase chain reaction (PCR) testing and admitted to the Department of Infectious Diseases, Tokyo Medical University Hachioji Medical Center, Tokyo, Japan, between 26 March 2021, and 1 November 2021. In Tokyo, it has been demonstrated that the α strain of SARS-CoV-2 was predominant in the fourth wave of COVID-19, and the δ strain was predominant in the fifth wave of COVID-19 [13]. Hence, this research period was divided into the fourth wave of COVID-19 (from 26 March to 20 June 2021) and the fifth wave of COVID-19 (from 21 June to 1 November 2021). An observational retrospective study was performed and clinical data were investigated from the medical records, including sex, age, and severity of COVID-19.

### 2.2. Quantitation of Serum MBL

Blood was collected from each patient into a serum-separating tube by venous puncture on the day of admission, centrifuged at 2000 rpm for 10 min, and the supernatant serum was collected to measure the level of serum MBL. The level of serum MBL (ng/mL) was measured by sandwich enzyme-linked immunosorbent assay (ELISA) according to the manufacturer’s instructions (Quantikine^®^ ELISA Human MBL; R&D Systems, Inc., 614 McKinley Place NE, Minneapolis, MN 55413, USA)

### 2.3. Determination of MBL2 Gene Variants

Peripheral blood was collected into ethylenediaminetetraacetic acid-containing tubes for the determination of *MBL2* gene polymorphisms. According to the manufacturer’s instructions, genomic DNA was extracted from blood samples using a QIAamp^®^ DNA Blood Mini Kit (QIAGEN K.K.-Japan, Tokyo, Japan). Polymorphisms in exon 1 of the *MBL2* gene were genotyped by PCR using sequence-specific primers.

PCR reactions were performed in a 50 µL volume using 0.5 µM of each primer (forward: 5′-TAGGACAGAGGGCATGCTC-3′; reverse: 5′-CAGGCAGTTTCCTCTGGAAGG-3′; product size: 349 base pairs (bps)). After the initial lysis of cells at 94 °C for 5 min, PCR reactions were run for 35 cycles of 30 s at 94 °C, 30 s at 60 °C, and 60 s at 72 °C, with a final extension at 72 °C for 5 min. The obtained PCR products (349 bp) were digested at 37 °C for 60 min with the restriction enzyme Ban I, according to the manufacturer’s protocol (Life Technologies, Carlsbad, CA, USA), which produced two fragments of 260 and 89 bp for the wild-type *MBL2* genotype (AA), three fragments of 349, 260, and 89 bp for the heterozygous *MBL2* genotype (AB), and an uncut fragment of 349 bp for the mutant homozygous *MBL2* genotype (BB). The digested products were loaded onto a 2% agarose gel and analyzed by electrophoresis [5]. 

### 2.4. Determination of COVID-19 Severity

COVID-19 severity was categorized into five stages, namely, mild, moderate, severe, critical, and death, according to the modified Guidelines on the Diagnosis and Treatment of Novel Coronavirus issued by the Ministry of Health, Labour, and Welfare, Japan (Guide 6.0 medical care of new coronavirus infection of the Ministry of Health, Labour and Welfare in Japan. 2 November 2021). Briefly, mild disease was defined as a lack of respiratory symptoms, no pulmonary radiological manifestations, and oxygen saturation levels (SpO_2_) ≥ 96%. Moderate disease was defined as mild respiratory symptoms, radiological evidence of pneumonia, and 93% < SpO_2_ < 96%. Severe disease was defined as SpO_2_ ≤ 93% and requiring oxygen support. Critical was defined as requiring a heart–lung machine or extracorporeal membrane oxygenation support for acute respiratory distress syndrome.

### 2.5. Statistical Analysis

Statistical analyses were performed using the statistical package for social science (SPSS) version 27 (IBM, Armonk, NY, USA). Statistical significance was considered to be a *p*-value of less than 0.05 for the results of all analyses. Kruskal–Wallis tests were used for continuous variables, and Fisher’s exact tests were used for binary variables. The Mann–Whitney U test was used for the comparison of averages. A scatter diagram and Spearman’s rank correlation coefficient were used to determine correlations. The Bonferroni correction for multiple testing was also applied. To identify predisposing factors of disease severity, binary logistic regression analysis was performed by calculating the odds ratios (ORs) at a confidence interval (CI) of 95%.

## 3. Results

### 3.1. Number of Patients, Age, and Serum MBL Levels

A total of 59 patients from the fourth wave and 49 patients from the fifth wave of PCR-confirmed COVID-19 were analyzed in this study. The largest proportion of patients were in their fifties (fourth wave: 15; fifth wave: 13), and there was a higher proportion of men (total number of patients: 108, men: 77, 67.6%) (Table 1). All patients were analyzed based on their serum MBL levels using ELISA. There was no significant association between serum MBL levels and age in both fourth and fifth waves (Figure 1). 

### 3.2. Each MBL Genotype and Age

The number of patients with each of the AA, AB, and BB genotypes was 41, 10, and 8 (fourth wave), and 26, 12, and 11 (fifth wave), respectively (Table 2). Table 2a shows the age of the patients with each MBL genotype in the fourth wave, and Table 2b shows the age of the patients with each MBL genotype in the fifth wave. There was no significant difference between each MBL genotype and age in both fourth and fifth waves (Kruskal–Wallis test, fourth wave: *p* = 0.11; fifth wave: *p* = 0.16).

### 3.3. Different COVID-19 Severities, Each MBL Genotype and Serum MBL Levels

Table 3 shows the comparison between different COVID-19 severities and each MBL genotype, and serum MBL levels in the fourth wave (Table 3a) and fifth wave (Table 3b). There was no significant difference in different COVID-19 severities and MBL genotypes in both fourth and fifth waves (Fisher’s exact test, fourth wave: *p* = 0.22, fifth wave: *p* = 0.63). There was no significant difference in different COVID-19 severities and serum MBL levels in both fourth and fifth waves (Kruskal–Wallis test, fourth wave: *p* = 0.42, fifth wave: *p* = 0.83).

### 3.4. Comparison between Each MBL Genotype and Serum MBL Levels

Figure 2 shows the serum MBL levels in patients with each MBL genotype in the fourth and fifth waves. In the fourth wave, patients with the AB or BB genotype tended to have significantly lower serum MBL levels than those with the AA genotype (Mann–Whitney U test, AA–AB: *p* < 0.01, AA–BB: *p* < 0.01). In the fifth wave, patients with the BB genotype had significantly lower serum MBL levels than those with the AA genotype (Mann–Whitney U test, AA–AB: *p* < 0.07, AA–BB: *p* < 0.03). 

### 3.5. Comparison between Allele Frequency in the Fourth and Fifth Waves, and the East Asian Database

We used an East Asian database, namely, the Japanese Multi Omics Reference Panel [14], as a web-based database for the comparison of allele frequencies (Table 4). In the fourth wave, the B-allele was found to be equal to or slightly higher in our cohort than in the East Asian database (0.22 versus 0.18, OR: 1.13; 95% CI: 0.82–1.57; *p* = 0.56). In the fifth wave, B-allele frequency was found to be significantly higher in our cohort than in the East Asian database (0.35 versus 0.18, OR: 2.13; 95% CI: 1.69–2.69; *p* < 0.001).

### 3.6. Binary Logistic Regression Analysis to Identify Predisposing Factors for More Than a Mild Level of COVID-19 Severity and Death from COVID-19

We performed binary logistic regression analysis to identify predisposing factors for a more than mild level of COVID-19 severity (Table 5), and death from COVID-19 (Table 6). Table 5 shows that the OR of age was 1.06. In other words, older patients have a higher risk of a more than mild level of COVID-19 severity (OR: 1.06; 95% CI: 1.03–1.08; *p* < 0.001). Table 6 shows that the OR of age was 1.11, and the OR of the BB genotype was 5.66. In other words, older patients and those with the BB genotype have a higher risk of death from COVID-19 (for age, OR: 1.11; 95% CI: 1.06–1.17; *p* < 0.001; for the BB genotype, OR: 5.66; 95% CI: 1.10–29.05; *p* = 0.04). These results indicate that the BB genotype might be significantly associated with death from COVID-19.

## 4. Discussion

A total of 108 patients were analyzed in this study, with a higher proportion of men (77 men, 67.6%) in the sample [15]. There was no significant association between serum MBL levels and age. In normal Japanese individuals, a decrease in MBL with age has been demonstrated [16]. This discrepancy was due to the small number of only hospitalization cases analyzed. The *MBL2* genotype AA was the most common, followed by genotype AB and genotype BB, which was similar to other literature [10]. The *MBL2* genotype is independent of age, as it is stable over the lifetime; however, genotypes might affect longevity. Three variants in noncoding regions of MBL2 have been identified: H/L (nucleotide-550), X/Y (nucleotide-221), and P/Q (at +4 in the 5′ UTR). Including codons 52, 54, and 57, in total there are six *MBL2* variants, which result in seven common haplotypes. They correspond to high, intermediate, low, and null biologic activity of the protein [17,18]. Tomaiuolo et al. demonstrated that the frequency of the intermediate activity haplotype is significantly higher in centenarians [19]. 

There was no significant difference in COVID-19 severities, MBL genotypes, and serum MBL levels. To the best of our knowledge, there have been no previous Japanese reports on the allele frequencies of the B variant of the *MBL2* gene at codon 54 in COVID-19 patients. The B-allele frequency in our study was equal to or slightly higher (fourth wave) or significantly higher (fifth wave) than that of the East Asian database. Medetalibeyoglu et al. [10] demonstrated that patients with the MBL B variant tend to have a lower level of serum MBL and a higher risk of a more severe clinical course of COVID-19. Genetic polymorphisms in the *MBL2* gene can interfere with MBL protein assembly, leading to a lower serum level of MBL. The *MBL2* genotype BB results in the overall absence of MBL protein whereas the *MBL2* AB genotype results in a 5- to 10-fold decrease in MBL protein compared to the *MBL2* AA genotype [10]. Our study also demonstrated that the B variant is associated with a lower level of serum MBL; however, there was no significant association between the level of serum MBL and disease severity. This discrepancy may be a result of the large number of patients that were analyzed by Medetalibeyoglu et al. [10]; they included a sample of 284 PCR-confirmed COVID-19 patients and 100 healthy controls. In our study, there were no control samples. However, Su et al. reported the *MBL2* genotype and serum MBL levels in 315 healthy controls in the same Asian race [20]. The *MBL2* genotype AA was the most common; the serum median level was 2257 ng/mL, similar to our data, and the *MBL2* genotype AB/BB was 2179 ng/mL, which was higher than our data indicated (Figure 2). This supports the association of the *MBL2* B allele with a lower level of serum MBL following infection by SARS-CoV-2. The *MBL2* B allele was demonstrated to be a risk factor for severe COVID-19 in some previous studies, and hence *MBL2* gene variants are thought to play a very important role in COVID-19 susceptibility and disease severity [21,22]. Speletas et al. [21] demonstrated that the presence of the MBL deficiency-causing B allele (*rs1800450*) was associated with an almost 2-fold increased risk of developing pneumonia and requiring hospitalization, suggesting its utility as a molecular predictor of disease severity in individuals with COVID-19. Tukek et al. [22] reported a COVID-19 family of five patients; the father had BB, the mother had AA, and the three children had the AB *MBL2* gene variant. The clinical course of the father was the most severe.

The main findings of this study were from the binary logistic regression analysis performed to identify predisposing factors for COVID-19 severity. We found that older patients have a significantly higher risk of developing a more than mild level of disease, which was similar to previous studies demonstrating that age is an important risk factor for disease severity [23,24]. The age-related decrease of physiological reserve in respiratory and other organ systems, and age-related reduction of CD4+ T cells, CD8+ T cells, and B cell functions, are said to weaken immune protection. However, aging is an important exacerbation factor in immune protection for all pathogens, not only SARS-CoV-2. It is important to find predisposing factors only for COVID-19 severity. In our study, regarding death from COVID-19, the BB genotype was found to have a stronger influence than age. Unlike previous clinical studies, our results quantitatively showed that the BB genotype might be associated with death from COVID-19. There are some healthy younger patients that follow a worse course while some morbid older patients follow a benign course. Genetic polymorphisms in the *MBL2* gene might have a stronger influence in these cases.

In severe COVID-19, acute respiratory distress syndrome is frequent and crucial. Many severe symptoms of COVID-19 have been reported, including cytokine storms, endothelial damage, and the progression of thromboses. Autopsies reveal arterial thrombosis and severe endothelial damage [25]. Cytokine storms lead to severe clinical manifestations of COVID-19, and impaired acquired immune responses and uncontrolled innate inflammatory responses may be associated with the mechanism of the cytokine storms in COVID-19 [26]. Endothelial damage can cause microvascular angiopathy and thrombosis. In human in vitro and animal studies, endothelial damage specifically activates the lectin pathway on the endothelial cell surface [27]. Coagulation/fibrinolytic abnormalities substantially affect the severity of COVID-19 [28]. Consistent with our hypothesis, individuals with the BB genotype might have a high level of inflammatory cytokines and D-dimers, and consequently, they might develop cytokine storms or thrombosis more easily, resulting in death. Further studies are needed to clarify these points.

Factors that contribute to the outcomes of COVID-19 remain unclear; however, MBL, which is important in innate immunity, might be the key to establishing new treatment strategies and developing novel therapeutic drugs. For diseases other than COVID-19, clinical trials, which include complement-targeting therapies, are performed for primary IgA nephropathy, which has no disease-specific treatment and might lead to kidney failure [11]. Recently, some papers have demonstrated that the administration of mannose-specific plant lectins may potentially prevent SARS-CoV-2 infection [29,30], and Takasumi et al. [31] demonstrated that a novel complement inhibitor—the human form of small mannose-binding lectin-associated protein factor H—targets both the lectin and alternative complement pathway. Rambaldi et al. [25] demonstrated that Narsoplimab, a fully human immunoglobulin gamma 4 (IgG4) monoclonal antibody against MASPs, inhibits lectin pathway activation and has anticoagulant effects. Narsoplimab treatment was associated with rapid and sustained reduction of circulating endothelial cells and inflammatory cytokines, it may be an effective treatment by reducing COVID-19-related endothelial cell damage and the resultant inflammation and thrombotic risk.

In the future, it is necessary to pay attention to not only MBL but also MASPs and ficolin. The MASP-2 MASP is a key enzyme in the lectin pathway of complement activation. Hyperactivation of MASP-2 by SARS-CoV-2 was found to contribute to aberrant complement activation, which led to worse lung injury and fatal prognosis [32]. Ficolin, together with MBL, belongs to the family of pattern-recognition molecules specific to the lectin pathway. These pattern-recognition molecules form complexes with MASPs. The influence of the lectin pathway on several infectious pathogens including SARS-CoV-2 has been reported [33].

## 5. Limitations

This study has several limitations. First, the patients enrolled in this study were from a specific part of Japan, namely, Tokyo (the population of Tokyo is about 14 million), and its suburbs (the population of Hachioji is about 580,000), and the sample size was too small to reach any clear conclusions. Second, there were no control samples for the serum MBL level. Third, the time of measurement of the serum MBL level after developing clinical symptoms was not consistent. For some patients, we analyzed the level of serum MBL at two time points, i.e., the acute phase and the convalescence phase (data not shown). Serum MBL levels were not constant over time, and for the patients who died, we were only able to measure the serum MBL level at one time point because the patient’s clinical course rapidly deteriorated. We measured the level of serum MBL only on the day of admission, and hence there was the possibility that the level fluctuated during the clinical course. Fourth, although nationwide vaccination against SARS-CoV-2 started in February 2021, the vaccination status of the subjects was not determined in this study. Fifth, in Tokyo, it has been demonstrated that the SARS-CoV-2 α strain was predominant in the fourth wave and the δ strain was predominant in the fifth wave; however, we did not analyze the strain of SARS-CoV-2 infection of the patients. Sixth, in the binary logistic regression analysis to identify predisposing factors for severity, the sample size was too small to include other factors, such as underlying diseases or comorbidities. Finally, the immunological function of MBL is affected not only by *rs1800450* (B variant, codon 54) but also by *rs5030737* C>T (D variant, codon 52) and *rs1800451* G>A (C variant, codon 57). In addition to variants in exon 1, there are also the gene promoter variants *rs11003125* (H/L) and *rs7096206* (Y/X) [34]. Monticelli et al. [35] also demonstrated that the *rs1800451* (Gly57Glu) allele frequency was associated with death from COVID-19. Hence, there are many genetic factors associated with susceptibility to COVID-19 infection and disease severity.

## 6. Conclusions

This was the first report, to our knowledge, demonstrating that the BB genotype might be associated with COVID-19-related mortality. Our results quantitatively showed that the BB genotype might be associated with death from COVID-19. However, our sample size was too small to further confirm whether the BB genotype is a predisposing factor for COVID-19-related mortality—analysis of a large population from all over Japan is needed. Further research on genetic factors and disease severity is warranted, and additional powerful and larger studies are needed for the establishment of new treatment strategies, such as complement-targeting therapies and therapeutic drugs, in the future.

## Figures and Tables

**Figure 1 life-13-00382-f001:**
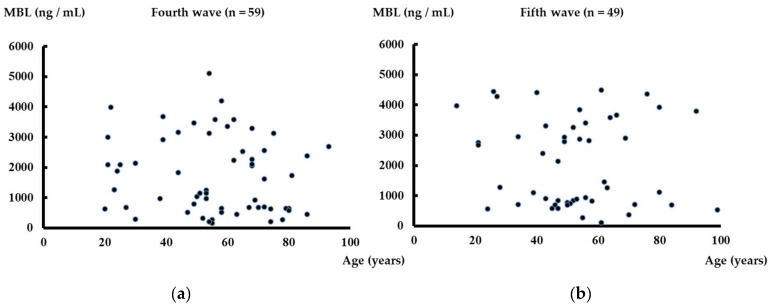
Serum MBL levels and age. There was no significant association between serum MBL levels and age in fourth wave (**a**). There was no significant association between serum MBL levels and age in fifth wave (**b**).

**Figure 2 life-13-00382-f002:**
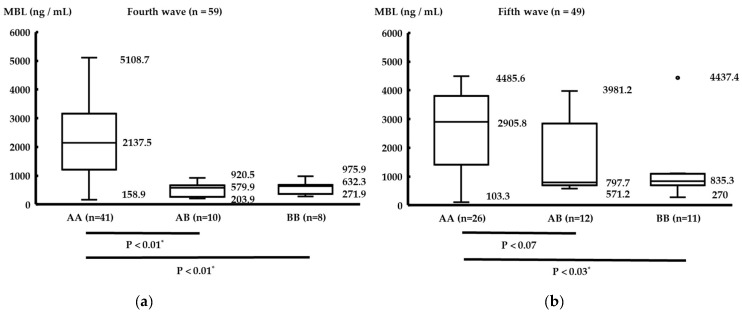
Comparison between each MBL genotype and serum MBL levels. The numbers indicate median (range). * Significant difference. In the fourth wave, patients with the AB or BB genotype tended to have significantly lower serum MBL levels than those with the AA genotype (**a**). In the fifth wave, patients with the BB genotype had significantly lower serum MBL levels than those with the AA genotype (**b**).

**Table 1 life-13-00382-t001:** No. patients and age of patients included in the study.

Age (y)	10–19	20–29	30–39	40–49	50–59	60–69	70–79	80–89	90–99	Total (Men/%)
Fourth wave	0	8 (7)	5 (5)	5 (2)	15 (9)	11 (9)	9 (6)	5 (2)	1 (0)	59 (40/67.8%)
Fifth wave	1 (0)	6 (3)	3 (3)	11 (6)	13 (10)	7 (6)	3 (2)	3 (2)	2 (1)	49 (33/67.3%)
Total (men)	1 (0)	14 (10)	8 (8)	16 (8)	28 (19)	18 (15)	12 (8)	8 (4)	3 (1)	108 (77/67.6 %)

**Table 2 life-13-00382-t002:** Comparison between each MBL genotype and age.

(a) Fourth wave (n = 59)					
	AA	AB	BB	Test of significance	*p*-value
	41	10	8		
Age Median(range)	54 (20–93)	70.5 (30–86)	62.5 (27–80)	Kruskal–Wallis	0.11
(b) Fifth wave (n = 49)					
	AA	AB	BB	Test of significance	*p*-value
	26	12	11		
Age Median(range)	55 (21–99)	45.5 (14–84)	50 (26–80)	Kruskal–Wallis	0.16

**Table 3 life-13-00382-t003:** Comparison between different severities and each MBL genotype (Fisher’s exact test), and serum MBL levels (Kruskal–Wallis test).

(a) Fourth wave (n = 59)							
	Mild	Moderate	Severe	Critical	Death	Test of significance	p Value
AA	18	6	11	2	4		
AB	3	1	4	0	2	Fisher’s exact	0.22
BB	2	2	0	1	3		
serum MBL levels (ng/mL)							
Median(range)	1254.6 (158.9–3988.7)	975.5 (289.2–3680.9)	3170.7 (641.9–5108.7)	2106.0 (975.9–2264.4)	686 (271.9–3132.2)	Kruskal–Wallis	0.42
(b) Fifth wave (n = 49)							
	Mild	Moderate	Severe	Critical	Death	Test of significance	p Value
AA	9	5	6	3	3		
AB	7	1	0	1	3	Fisher’s exact	0.63
BB	6	1	1	1	2		
serum MBL levels (ng/mL)							
Median(range)	1521.9 (270–4437.4)	2783.8 (1092.7–4407.9)	1257.1 (103.3–3590.0)	1462.8 (574.8–3840.7)	1026.0 (537–4485.6)	Kruskal–Wallis	0.83

**Table 4 life-13-00382-t004:** Comparison between allele frequency in the fourth and fifth waves, and the East Asian database.

Allele	A	B	OR	95% CI	*p* Value
Fourth wave	0.78	0.22			
Fifth wave	0.65	0.35	1.13	0.82–1.57	0.56
East Asian Database	0.82	0.18	2.13	1.69–2.69	<0.001 *^1^

*p* values indicate differences between the A and B alleles in the fourth and fifth waves and the East Asian database. *^1^
*p*-value between the B allele in the fifth wave and the East Asian database.

**Table 5 life-13-00382-t005:** Binary logistic regression analysis to identify predisposing factors for more than a mild level of COVID-19 severity.

	*p*-Value	OR	95% CI
Sex	0.16	0.51	0.20–1.23
Age	<0.001 *	1.06	1.03–1.08
AA genotype	0.89		
AB genotype	0.69	0.80	0.26–2.44
BB genotype	0.72	0.82	0.27–2.51

Model χ^2^: *p* < 0.001; Hosmer & Lemeshow: *p* = 0.08; * Significant difference.

**Table 6 life-13-00382-t006:** Binary logistic regression analysis to identify predisposing factors for death from COVID-19.

	*p*-Value	OR	95% CI
Sex	0.92	0.93	0.23–3.82
Age	<0.001 *	1.11	1.06–1.17
AA genotype	0.10		
AB genotype	0.23	2.63	0.55–12.59
BB genotype	0.04 *	5.66	1.10–29.05

Model χ^2^: *p* < 0.001; Hosmer & Lemeshow: *p* = 0.54; * Significant difference.

## Data Availability

Not applicable.

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
