# Peer review of "Association of the Mannose-Binding Lectin 2 BB Genotype with COVID-19-Related Mortality"

_life, 2023, doi:10.3390/life13020382_

Round 1

Reviewer 1 Report

The author has done a good effort but it needs some technical corrections. The manuscript needs an English proofreading. The manuscript can be processed further after addressing the following comments:

1.     The title needs to be rewritten. Maybe it can be “Association of mannose-binding lectin 2 BB genotype 2 with COVID-19 related mortality”.

2.     The first sentence in the summary and abstract is repetitive. It should be corrected.

3.     Line 32-33: the authors should mention here which type of sample they collected and analyzed for MBL and using which technique.

4.     Authors need to provide more results in the abstract.

5.     Line 66-75: there are no references cited in the text. But author has referenced some studies from Japan. Please cite here the respective studies.

6.     In section 2.1.: Please also mention here the Ethical approval.

7.     Line 86-89: waves of COVID-19.

8.     Line 137-138: The sentence needs to be revised.

9.     Figures: I suggest not merging tables with the figures. Authors can keep figures only and remove the tables.

10.  Line 200: “To the best of our knowledge”.

11.  The conclusion section should be one paragraph.

12.  Line 274: remove “7. Patents”.

13.  Line 291-310: Remove these from here.

14.  For tables and references, authors are advised to follow the publisher’s guidelines.

15.  There are total of 18 citations at the end, but I could see only 16 in-text citations. Authors are advised to recheck.

16.  In the discussion section, there are only 6 references, authors are advised to cite more studies and discuss their results with other studies.

17.  Line 260-265: Font size is different.

18.  Line 284: Authors acknowledge the English proofreading services, but in my opinion, it still requires corrections.

Author Response

Reviewer 1

Thank you for your comments.

I revised my paper as you pointed out.

The author has done a good effort but it needs some technical corrections. The manuscript needs an English proofreading. The manuscript can be processed further after addressing the following comments:

  1. The title needs to be rewritten. Maybe it can be “Association of mannose-binding lectin 2 BB genotype 2 with COVID-19 related mortality”. âž¡ I changed the title.
  2. The first sentence in the summary and abstract is repetitive. It should be corrected.

âž¡ I revised the summary and abstract.

  1. Line 32-33: the authors should mention here which type of sample they collected and analyzed for MBL and using which technique. âž¡ I mentioned about samples and technique.
  2. Authors need to provide more results in the abstract. âž¡ I revised the abstract.
  3. Line 66-75: there are no references cited in the text. But author has referenced some studies from Japan. Please cite here the respective studies. âž¡ I revised the introduction.
  4. In section 2.1.: Please also mention here the Ethical approval. âž¡ I mentioned the Ethical approval as you pointed out.
  5. Line 86-89: waves of COVID-19. âž¡ I revised ‘wave’ to ‘waves’.
  6. Line 137-138: The sentence needs to be revised. âž¡ I revised the sentence as you pointed out.
  7. Figures: I suggest not merging tables with the figures. Authors can keep figures only and remove the tables. âž¡ This time, I revised all figures and tables.
  8. Line 200: “To the best of our knowledge”. âž¡ I revised the sentence as you pointed out.
  9. The conclusion section should be one paragraph. âž¡ I revised one paragraph as the conclusion section.
  10. Line 274: remove “7. Patents”. âž¡ I removed “7. Patents”.
  11. Line 291-310: Remove these from here. âž¡ I removed these sentences.
  12. For tables and references, authors are advised to follow the publisher’s guidelines. âž¡ I revised tables and references according to the publisher’s guideline.
  13. There are total of 18 citations at the end, but I could see only 16 in-text citations. Authors are advised to recheck. âž¡ I revised the references according to your comments.
  14. In the discussion section, there are only 6 references, authors are advised to cite more studies and discuss their results with other studies. âž¡ I revised the discussion section according to your comments.
  15. Line 260-265: Font size is different. âž¡ I aligned font size.
  16. Line 284: Authors acknowledge the English proofreading services, but in my opinion, it still requires corrections. âž¡ I think the proofreading of contents is needed, I want to use the English proofreading last.

Thank you for your comments.

Yasuyo Kashiwagi

Department of Pediatrics and Adolescent Medicine, Tokyo Medical University, 6-7-1

Nishishinjuku, Shinjuku-ku, Tokyo 160-0023, Japan

E-mail: hoyohoyo@tokyo-med.ac.jp

Reviewer 2 Report

The paper is well written and provides the preliminary study.

However, the sample size taken for the study is too small and no controls are taken. 

Apart from that the authors themselves reported so many limitations in their article which can not be ignored. 

Therefore, I can not recommend the article for publication in the present form. 

Thus, article is rejected for publication.

Author Response

Reviewer 2

The paper is well written and provides the preliminary study.

However, the sample size taken for the study is too small and no controls are taken. 

Apart from that the authors themselves reported so many limitations in their article which can not be ignored. 

Therefore, I can not recommend the article for publication in the present form. 

Thus, article is rejected for publication.

➡Thank you for your comments.

I revised my paper.

Yasuyo Kashiwagi

Department of Pediatrics and Adolescent Medicine, Tokyo Medical University, 6-7-1

Nishishinjuku, Shinjuku-ku, Tokyo 160-0023, Japan

E-mail: hoyohoyo@tokyo-med.ac.jp

Reviewer 3 Report

Although this investigation has several major limitations (some of them were well defined by Authors themselves), the data presented are interesting.  Results given here suggest that B/B MBL2 genotype may predict mortality from COVID-19. However, figures (especially these “table-like”) are far too small to analyse them with no magnification. Moreover, data presented in fig. 2 seem not to be very valid, as genotype is independent of age and vice versa (there is no greater influence of MBL2 genotype on lifetime, at least in general population, although some reports suggested that heterozygosity might be beneficial). Another problem is the quality of writing. For example:

-        Mannose-binding lectin (MBL) is important in first-line immune defenses against several 25 important pathogens including coronavirus disease 2019 (COVID-19) – COVID-19 is not pathogen but disease (pathogen is SARS-CoV-2)

-        Mannose-binding lectin (MBL) was the first molecule identified to be involved in 41 innate immunity….. – surely, MBL was not the first

-        MBL insufficiency is caused by 3 structural variants in exon 1, resulting in MBL dysfunction and susceptibility to various infectious and autoimmune diseases, such as rheumatoid arthritis [2], Kawasaki disease [3,4], recurrent vulvovaginal candidiasis [5], respiratory distress syndrome in preterm infants [6], hepatitis B infection [7], severe acute respiratory syndrome coronavirus infection [8], etc. However, whether these MBL2 gene variants are associated with susceptibility to various diseases remain unknown and controversial. – The second sentence practically contradicts the first

-        Figure 3 shows that serum MBL levels of patients with different disease severities (Figure 3a), and number of patients with each MBL genotype and disease severity (Figure 3b) – “that” is not necessary (the same for “Figure 4 shows…..”)

Author Response

Reviewer 3

Thank you for your comments.

I revised my paper as you pointed out.

Although this investigation has several major limitations (some of them were well defined by Authors themselves), the data presented are interesting.  Results given here suggest that B/B MBL2 genotype may predict mortality from COVID-19. However, figures (especially these “table-like”) are far too small to analyse them with no magnification. âž¡This time, I revised all figures and tables.

Moreover, data presented in fig. 2 seem not to be very valid, as genotype is independent of age and vice versa (there is no greater influence of MBL2 genotype on lifetime, at least in general population, although some reports suggested that heterozygosity might be beneficial).

âž¡ I added these sentences in conclusion section, “MBL2 genotype is independent of age, there is no greater influence of MBL2 genotype on lifetime. However, to further confirm whether the BB genotype is a predisposing factor for COVID-19 related mortality, analysis of a large population from all over Japan is needed.”

Another problem is the quality of writing. For example:

-        Mannose-binding lectin (MBL) is important in first-line immune defenses against several 25 important pathogens including coronavirus disease 2019 (COVID-19) – COVID-19 is not pathogen but disease (pathogen is SARS-CoV-2) âž¡ I am sorry, I revised these sentences as you pointed out.

-        Mannose-binding lectin (MBL) was the first molecule identified to be involved in 41 innate immunity….. – surely, MBL was not the firstâž¡ I revised introduction section.

-        MBL insufficiency is caused by 3 structural variants in exon 1, resulting in MBL dysfunction and susceptibility to various infectious and autoimmune diseases, such as rheumatoid arthritis [2], Kawasaki disease [3,4], recurrent vulvovaginal candidiasis [5], respiratory distress syndrome in preterm infants [6], hepatitis B infection [7], severe acute respiratory syndrome coronavirus infection [8], etc. However, whether these MBL2 gene variants are associated with susceptibility to various diseases remain unknown and controversial. – The second sentence practically contradicts the firstâž¡ I revised introduction section.

-        Figure 3 shows that serum MBL levels of patients with different disease severities (Figure 3a), and number of patients with each MBL genotype and disease severity (Figure 3b) – “that” is not necessary (the same for “Figure 4 shows…..”) âž¡This time, I revised all figures and tables.

Thank you for your comments.

Yasuyo Kashiwagi

Department of Pediatrics and Adolescent Medicine, Tokyo Medical University, 6-7-1

Nishishinjuku, Shinjuku-ku, Tokyo 160-0023, Japan

E-mail: hoyohoyo@tokyo-med.ac.jp

Round 2

Reviewer 2 Report

Kindly work on the paper to reduce the limitations. 

Author Response

Reviewer 2

Thank you for your comments.

I revised my paper as you pointed out.

Kindly work on the paper to reduce the limitations. 

âž¡I tried to reduce the limitation, however, there are many limitations, I found it impossible to address reviewer’s comments. 

This time, we used the English editing service.

References added are in red.

Thank you for your comments.

Yours sincerely,

Yasuyo Kashiwagi, M.D., Ph.D.

Department of Pediatrics and Adolescent Medicine,
Tokyo Medical University
6-7-1 Nishishinjuku,
Shinjuku-ku, Tokyo 160-0023, Japan
e-mail: hoyohoyo@tokyo-med.ac.jp

Reviewer 3 Report

The Authors positively responded majority of my querries, however there are still two problems to be solved:

-        Authors probably misunderstood my suggestion concerning possible interplay between MBL2 genotype and lifetime as they wrote in revised manuscript: MBL2 genotype is independent of age, there is no greater influence of MBL2 genotype on lifetime (lines 296-297 and 367-368).  Indeed, as I suggested, MBL2 genotype is independent of age (as it stable during lifetime), however genotype might affect the lifetime (please see paper by Tomaiuolo et al., Aging Cell 2012, 11, 394-400). That’s not the same.

-        Please explain statistical analysis for table 4 (Fisher’s exact is not suitable for comparison of average (median or mean) concentrations between two independent groups – it is used rather for 2x2 table and comparison of frequency). As MBL concentrations are usually not normally distributed, the best solution seems Mann-Whitney

Minor point: line 45: SARS-CoV-2 instead of SARS-Cov-2

Author Response

Reviewer 3

Thank you for your comments.

I revised my paper as you pointed out.

-        Authors probably misunderstood my suggestion concerning possible interplay between MBL2 genotype and lifetime as they wrote in revised manuscript: MBL2 genotype is independent of age, there is no greater influence of MBL2 genotype on lifetime (lines 296-297 and 367-368).  Indeed, as I suggested, MBL2 genotype is independent of age (as it stable during lifetime), however genotype might affect the lifetime (please see paper by Tomaiuolo et al., Aging Cell 2012, 11, 394-400). That’s not the same.

➡ You are right, I revised my paper (lines 274-282).

-        Please explain statistical analysis for table 4 (Fisher’s exact is not suitable for comparison of average (median or mean) concentrations between two independent groups – it is used rather for 2x2 table and comparison of frequency). As MBL concentrations are usually not normally distributed, the best solution seems Mann-Whitney

➡ Table 4 was rewritten as Figure 2 using Mann-Whitney U test (lines 208-216) as you pointed out.

Minor point: line 45: SARS-CoV-2 instead of SARS-Cov-2

➡ I rewrote SARS-CoV-2.

This time, we used the English editing service.

References added are in red.

Thank you for your comments.

Yours sincerely,

Yasuyo Kashiwagi, M.D., Ph.D.

Department of Pediatrics and Adolescent Medicine,
Tokyo Medical University
6-7-1 Nishishinjuku,
Shinjuku-ku, Tokyo 160-0023, Japan
e-mail: hoyohoyo@tokyo-med.ac.jp
